# Spoken Expressive Vocabulary in 2-Year-Old Children with Hearing Loss: A Community Study

**DOI:** 10.3390/children10071223

**Published:** 2023-07-14

**Authors:** Peter Carew, Daisy A. Shepherd, Libby Smith, Tegan Howell, Michelle Lin, Edith L. Bavin, Sheena Reilly, Melissa Wake, Valerie Sung

**Affiliations:** 1Murdoch Children’s Research Institute, Royal Children’s Hospital, Parkville, VIC 3052, Australia; pcarew@unimelb.edu.au (P.C.); daisy.shepherd@mcri.edu.au (D.A.S.); libby.smith@mcri.edu.au (L.S.); tegan.howell@mcri.edu.au (T.H.); michelle.lin@mh.org.au (M.L.); e.bavin@latrobe.edu.au (E.L.B.); s.reilly@griffith.edu.au (S.R.); melissa.wake@mcri.edu.au (M.W.); 2Department of Audiology and Speech Pathology, The University of Melbourne, Parkville, VIC 3010, Australia; 3Department of Paediatrics, The University of Melbourne, Parkville, VIC 3010, Australia; 4School of Psychology and Public Health, La Trobe University, Bundoora, VIC 3086, Australia; 5Menzies Health Institute Queensland, Griffith University, Gold Coast, QLD 4222, Australia; 6Liggins Institute, University of Auckland, Auckland 1023, New Zealand; 7Centre for Community Child Health, Royal Children’s Hospital, Parkville, VIC 3052, Australia

**Keywords:** language, unilateral hearing loss, bilateral hearing loss, intervention

## Abstract

Through a cross-sectional community study of 2044 children aged 2 years, we (1) examine the impact of hearing loss on early spoken expressive vocabulary outcomes and (2) investigate how early intervention-related factors impact expressive vocabulary outcomes in children with hearing loss predominantly identified through universal newborn hearing screening. We used validated parent/caregiver-reported checklists from two longitudinal cohorts (302 children with unilateral or bilateral hearing loss, 1742 children without hearing loss) representing the same population in Victoria, Australia. The impact of hearing loss and amplification-related factors on vocabulary was estimated using g-computation and multivariable linear regression. Children with versus without hearing loss had poorer expressive vocabulary scores, with mean scores for bilateral loss 0.5 (mild loss) to 0.9 (profound loss) standard deviations lower and for unilateral loss marginally (0.1 to 0.3 standard deviations) lower. For children with hearing loss, early intervention and amplification by 3 months, rather than by 6 months or older, resulted in higher expressive vocabulary scores. Children with hearing loss demonstrated delayed spoken expressive vocabulary despite whole-state systems of early detection and intervention. Our findings align with calls to achieve a 1-2-3 month timeline for early hearing detection and intervention benchmarks for screening, identification, and intervention.

## 1. Introduction

Universal newborn hearing screening (UNHS) effectively reduces the median age of congenital hearing loss detection [1,2]. This enables earlier diagnosis, hearing amplification, and intervention for deaf and hard-of-hearing (DHH) children, with improved language outcomes reported at school age compared to those detected or amplified later [3,4]. However, these improved language outcomes remain poorer than normative expectations and population means [5]. Therefore, we should continue to seek modifiable factors that may improve early language outcomes for DHH children, over and above well-established early detection, to reduce this gap between children with and without hearing loss to the greatest extent possible.

There are relatively few reports of the spoken language outcomes of young DHH children. Early intervention (EI)-based studies show mixed results, either demonstrating children can achieve age-appropriate speech/language outcomes by three years of age [6] or age-appropriate skills at 12–18 months that fall below expectations by age 3 years [7,8]. A study aggregating 12 United States (US) states’ EI-based data on children with bilateral hearing loss found parent-reported expressive vocabulary scores were more than one standard deviation poorer than expected by age 2 years [9]. Significantly better scores were observed in children who met Early Hearing Detection and Intervention (EHDI) guidelines (screen by 1 month, diagnose by 3 months, enroll in EI by 6 months of age) compared to those who did not. Consistent with previous reports [10], better language was observed for children with milder losses, fewer comorbidities, and higher maternal education. However, this may represent a best-case scenario, as it included outcomes only from children actively engaged in EI services and already amplified, and the sample appeared to be more advantaged and to have fewer disabilities than the general DHH population. Thus, these findings may not reflect the broader population of children with any degree of hearing loss impacting one or two ears.

While some catch-up is possible [11], poorer early vocabulary is an important predictor of later language skills [12,13], often presaging persistent communication and educational difficulties. As EHDI continues to evolve, so do children’s outcomes [14]. To inform changes in early management, we need to understand what influences very early vocabulary outcomes in today’s DHH children. Research needs to include those who are not amplified, are not accessing EI, and/or are accessing different interventions, and include appropriate same-population comparisons.

Comparing two community cohorts of 2-year-old children with and without hearing loss derived from the same Australian state, this study aimed to (1) describe the impact on spoken expressive vocabulary of all degrees of unilateral and bilateral hearing loss and (2) describe intervention-related factors (hearing amplification device use, age at first hearing amplification, and age at first enrollment with EI services) for DHH children with unilateral and bilateral losses and quantify their impact on expressive vocabulary for DHH children with bilateral hearing loss.

## 2. Materials and Methods

### 2.1. Study Design and Participants

This was a cross-sectional study of spoken expressive vocabulary in 2-year-old children from two longitudinal cohorts: DHH children from the Victorian Childhood Hearing Longitudinal Databank (VicCHILD) [15], and children without hearing loss from the Early Language in Victoria Study (ELVS) [16], both born or residing in the state of Victoria (population 6.7 million), Australia. Combining these two cohorts formed a single community sample enriched for hearing loss that represented expressive vocabulary skills across the full range of hearing. The studies were approved by the Ethics Committees of The Royal Children’s Hospital (VicCHILD and ELVS) and La Trobe University (ELVS), with parents/caregivers having provided written informed consent.

### 2.2. Participants with Permanent Hearing Loss 

VicCHILD is a population-level data repository for over 1100 children with any degree or type of permanent hearing loss. All children identified with hearing loss through Victoria’s UNHS program (99% uptake, 1.8% loss to follow-up) are invited to participate, as are children who attend the Royal Children’s Hospital Caring for Hearing In Children Clinic for congenital or late-onset hearing loss. Most VicCHILD participants are under one year of age at enrollment. All participants also have access to government-supported hearing amplification and EI programs. Data are collected longitudinally with repeated measures as the child grows. Further information on VicCHILD’s methodology is detailed elsewhere [15]. This study included all VicCHILD participants born between 2013 and 2019, with hearing, demographic, early language, and service use data from the first two collection points (enrollment and age around 2 years). 

### 2.3. Participants without Known Permanent Hearing Loss

ELVS has documented the speech and language development of a community sample of children without known permanent hearing loss, developmental delays, or serious disabilities when recruited in 2003-4 at ages 8–10 months [17]. Over 1900 children were recruited from six of Victoria’s 31 metropolitan local government areas, selected to represent children from different socio-economic backgrounds, and have been followed in successive waves. Demographic and spoken vocabulary outcomes data from participants at around 2 years of age were used, when the participant retention rate was 91.1% [16]. Further information on ELVS’s methodology is detailed elsewhere [17].

### 2.4. Outcome

The primary outcome was parent-/caregiver-reported spoken expressive vocabulary at around 2 years, collected in both cohorts using closely related measures. ELVS used the 680-word vocabulary checklist in the MacArthur Bates Communicative Development Inventory (MCDI) Words and Sentences test [18], whereas VicCHILD used the 100-word checklist in the Sure Start Language Measure (SSLM) [19], a validated measure based on the MCDI: UK Short Form [20]. Both the MCDI and SSLM have mean standard scores of 100 (standard deviation of 15) and, when compared, show high reliability and concurrent validity [19]. The study included DHH children aged 18–30 months at assessment whose parent/caregiver completed the SSLM and children without hearing loss aged 23.5–25.5 months whose parent/caregiver completed the MCDI. 

The expressive vocabulary outcome was derived from standardized SSLM scores for both cohorts. For the ELVS cohort, the SSLM raw score was calculated for the 100 items in the MCDI common to the SSLM. For any combination of words, such as “sofa/couch” being a single item on the SSLM but two items on the MCDI, a single score was assigned in the SSLM raw score calculation if at least one of the words was selected on the MCDI. Based on this process, MCDI scores were converted to SSLM raw score equivalents, then to standardized scores based on age (in months) and sex.

### 2.5. Exposures

The exposure for aim 1 was hearing loss, defined as no hearing loss or a combination of the degree of loss (mild (21–40 dB)/moderate (41–60 dB)/severe (61–90 dB)/profound (>91 dB)) and presence in one/both ears (unilateral/bilateral). Degree of loss, calculated using three or four frequency averages, was obtained from UNHS records or parent/caregiver-supplied audiology results, classified using national decibel ranges for the affected ear (unilateral) or better hearing ear (bilateral) [21].

For aim 2, intervention-related factors were considered as separate exposures to explore the impact of intervening on single characteristics individually. Within the bilateral hearing loss cohort, exposures were hearing amplification status at survey completion (amplified vs. unamplified), frequency of hearing device use (never/rarely (no device or use <4 h/day), sometimes/often (4–8 h/day) or always (>8 h/day) derived from parent/caregiver-estimated hours of daily use at time of assessment), and age first enrolled into an EI program (≤3 months, 3.1–6 months, >6 months). Age at first hearing amplification fitting (≤3 months, 3.1–6 months, >6 months) was an exposure for children ever fitted with hearing amplification.

### 2.6. Potential Confounders

Directed acyclic graphs were developed to model assumptions about the causal structures (see Appendix A). For both aims, we identified possible demographic-related confounders as child sex, parent education level (completed at least undergraduate education: yes/no), primary language at home (English only: yes/no), and social disadvantage (measured using the Australian census-based Socio-Economic Indexes for Area (SEIFA), national mean 1000, SD 100; higher scores represent less disadvantage) [20]. Possible birth-related confounders were Neonatal Intensive Care Unit (NICU) admission and gestational age.

### 2.7. Statistical Analyses 

All analyses were completed in statistical software R version 4.0.2 [22] and conducted for children with complete data for the respective aim only.

#### 2.7.1. Aim 1

For aim 1, the samples of individuals with and without hearing loss were combined. We used a causal modeling approach to estimate the impact of hearing loss on expressive vocabulary at age 2 years, outlined as follows: Linear regression models were fitted to the data, modeling the outcome conditional on exposure, with three different models considered. Model 1 was an unadjusted model. Model 2 included demographic confounders as covariates, with Model 3 additionally including birth-related factors as covariates. The latter two models included interaction terms where appropriate (see Appendix B for further detail). Models 2 and 3 were fitted to consider different confounding adjustment sets and, additionally, the trade-off between potential bias from the exclusion of birth-related confounders (Model 2) and from sparse data in confounder substrata (Model 3). Estimates of the mean difference in standardized SSLM scores between hearing loss groups were obtained by standardizing (i.e., g-computation [23]) over the hearing loss sample to estimate the effect of hearing loss on SSLM score within the hearing loss population. Confidence intervals and standard errors were obtained via non-parametric bootstrapping. Estimates from Model 3 were interpreted as the primary analysis, which adjusted for demographic and birth-related potential confounders aiming to minimize confounding bias in causal effect estimates, with results under all models also provided. Further details on the statistical analysis methods can be found in Appendix B and Appendix A.

#### 2.7.2. Aim 2

For aim 2, hearing-related characteristics of individuals with any hearing loss were described. When estimating the impact of intervention-related factors on expressive vocabulary, we hypothesized the impact would be different for individuals with unilateral or bilateral loss. Also, considering the small sample size of individuals with unilateral hearing loss, we focused on the bilateral group when estimating this impact. Intervention-related factors were considered separate exposure variables and analyzed individually. Three regression models were again considered for each exposure, with the same covariates included as were used in aim 1 based on the same assumption around potential confounders and causal structures. Models 2 and 3 were also adjusted for bilateral hearing loss severity, as we considered this to be an influential factor in the expressive vocabulary of individuals. As with aim 1, Model 3 was interpreted as the primary analysis. For categorical variables with more than two levels, estimates of the mean difference in SSLM scores were calculated compared to a baseline level. We selected the baseline to reflect the “best case scenario”, i.e., earlier intervention, earlier age of detection.

## 3. Results

### 3.1. Sample Characteristics

The sample included 302 DHH children (75.5% of VicCHILD participants approached) and 1742 children from the ELVS cohort who were considered to not have permanent hearing loss (Figure 1). The mean age at assessment was similar between groups (25.4 months and 24.2 months, respectively), with more children without hearing loss being from an English-speaking-only household (Table 1). Children without hearing loss tended to be less disadvantaged than DHH peers (mean SEIFA index 1037.6 versus 1015.6), although both were above the national SEFIA mean of 1000. Of the DHH children, 209 (69.2%) had bilateral hearing loss, with a relatively equal distribution across degrees of loss.

### 3.2. Aim 1: Impact of Hearing Loss on Expressive Vocabulary 

Estimates from Model 3 are presented in Table 2 as the primary analysis, along with estimates from Models 1 and 2. Adjusted mean expressive vocabulary scores were estimated to be lower for DHH children (of any severity and in either ear) compared to children with no hearing loss (Table 2). For bilateral hearing loss, the impact became more substantial as the degree increased. Compared to no hearing loss, adjusted mean expressive vocabulary scores ranged from 7.3 points (or 0.5 of a standard deviation) lower for mild bilateral loss (95% CI −11.4 to −2.5, *p* < 0.01) to 13.5 points (0.9 standard deviations) lower for profound bilateral loss (95% CI −18.5 to −8.4, *p* < 0.01). For unilateral hearing losses, the impact was less substantial, with adjusted mean scores ranging between 1.5 and 4.4 points lower (0.1 and 0.3 of a standard deviation, respectively) (Table 2).

### 3.3. Aim 2: Impact of Intervention-Related Factors on Expressive Vocabulary in Children with Hearing Loss

Estimates from Model 3 are again presented, with the distribution of hearing loss under each exposure group presented in Appendix A. Two hundred and sixteen children (71.5% of 302 children from aim 1) used hearing amplification at survey completion, with 86.6% having bilateral hearing loss (187/216). Of the 22 children with unaided bilateral hearing loss, 81.8% had mild loss (Figure 2). For unilateral hearing loss, 31.2% were fitted with hearing amplification at the time of the survey (29/93) (Table 3). Unlike bilateral losses, there was no obvious relationship between the degree of unilateral loss and the distribution of intervention-related factors. For children with bilateral hearing loss, adjusted mean expressive vocabulary scores were lower if hearing amplification was used at time of assessment compared to scores of children without hearing amplification (adjusted mean difference 5.4 points, 95% CI −2.5 to 13.2 points, *p* = 0.18) (Table 4).

Reflective of a well-functioning UNHS environment, the median age at diagnosis was 1.0 month for children with any hearing loss. Most children received hearing amplification early (median age 3 months; Table 3), with fitting generally earlier if the degree of bilateral hearing loss was greater (Figure 2), whereas 46% of children with mild bilateral loss were fitted after age 6 months. Hearing amplification at 3 months or younger was associated with higher mean expressive vocabulary scores compared to first amplification when older than 3 months for individuals with bilateral hearing loss (adjusted mean difference of 8.6 points, 95% CI 3.1 to 14.1, *p* < 0.01 and 4.3 points, 95% CI −1.7 to 10.3, *p* = 0.16 for ages 3.1 to 6 months and >6 months, respectively; Table 4).

The relationship between hearing amplification use and degree of loss appeared stronger for children with bilateral loss versus unilateral loss; children rated as always wearing their hearing device tended to have greater degrees of bilateral loss compared to other use categories (Figure 2). Higher expressive vocabulary scores were estimated for children with bilateral hearing loss who never/rarely (rated to have <4 h daily average use or no hearing device fitted) wore hearing amplification, followed by those who always wore (>8 h daily average use), with those who sometimes/often (4–8 h daily average use) wore hearing amplification demonstrating the lowest expressive vocabulary scores (Table 4).

Most DHH children (with either unilateral or bilateral hearing loss) were enrolled in EI services early (median age 6 months) and were enrolled with a service at survey completion (Table 3). Children enrolled at older ages tended to have mild or moderate bilateral hearing loss (Figure 2). Mean SSLM scores for children with bilateral hearing loss were higher if enrolment was in the first 3 months of life compared to older ages (adjusted mean difference of 5.4 points, 95% CI −0.1 to 11.0, *p* = 0.05, and 10.0 points, 95% CI 4.2 to 15.7, *p* < 0.01 for enrolment at ages 3.1–6 months and >6 months, respectively, Table 4).

## 4. Discussion

### 4.1. Principal Findings

Despite Victoria’s sophisticated early identification and intervention systems reaching essentially all children, the spoken expressive vocabulary of 2-year-old DHH children lagged behind those without hearing loss. Spoken expressive vocabulary of children with bilateral losses was increasingly impacted as the degree of hearing loss increased, ranging from 0.5 to 0.9 standard deviations below expected levels after adjustment for potential demographic and birth-related confounders. Children with unilateral losses had an expressive vocabulary closer to (yet, on average, still poorer than) children without hearing loss, without a clear relationship between outcome and hearing loss severity.

Importantly, we demonstrated enrollment in EI by 3 months of age resulted in higher spoken expressive vocabulary scores. A similar pattern towards higher expressive vocabulary scores was also observed in the presence of earlier amplification. This may provide support for the narrower 1-2-3 month alternative timeline to the 1-3-6 EHDI indicators [24]. However, the association between hearing amplification use and language outcomes was not straightforward. We describe a U-shaped relationship where children rated as either “never/rarely” or “always” using hearing amplification showed greater expressive vocabulary scores than those children rated as “sometimes/often” using their hearing device(s).

### 4.2. Strengths of the Study

Our 96% UNHS-identified community cohort represents the common early diagnosis pathway occurring in countries with well-resourced EHDI systems. By including children with any degree of hearing loss in either ear and those with unamplified hearing losses, this enables a more generalizable estimate of expressive vocabulary performance than previous studies of early intervention cohorts (which excluded unilateral losses [9] or only included children with hearing amplification [9] or who had very low birthweight [25]). Despite sample sizes restricting the precision of estimates, our causal modeling approach is more flexible than other analysis approaches used elsewhere [9] (see Appendix B).

### 4.3. Limitations

Our statistical approach attempted to minimize potential confounding biases. When interpreting the results, we acknowledge the trade-off between sparse data in confounder substrata (under Model 3) and the potential for additional unmeasured confounding bias (under Model 2). Therefore, results are presented under all models to allow the reader to interpret the estimates while acknowledging these potential limitations. Unmeasured residual confounding remains possible, such as nonverbal IQ and differences between our hearing and hearing loss groups with respect to the inclusion of children with developmental delays. Limited by sample size, we analyzed exposures separately for aim 2. This meant we were unable to consider the potential combined effects of these exposures.

Our study was limited to examining spoken vocabulary as a measure of expressive language in DHH children, acknowledging that sign language is an important form of communication for DHH children. Among our VicCHILD participant children, less than 3% were reported to use Australian sign language (Auslan) as the predominant mode of communication. Moreover, there are no standardized or validated tools to measure sign language outcomes in preschool-aged children.

Children with hearing loss were drawn from our population-level data repository, with 60% of participant parents/caregivers reporting their child had no additional special health need or medical diagnosis. This is in keeping with other studies reporting that around 40% of DHH children have concomitant medical comorbidities; however, our study did not collect adequate information about medical comorbidities to determine whether this could be a factor influencing language outcomes.

For aim 2, we assumed the impact of intervention-related factors on expressive vocabulary would be different between individuals with unilateral or bilateral loss and therefore presented the estimated impact for bilateral losses only. A secondary analysis was conducted to estimate the impact for individuals with any permanent hearing loss—unilateral or bilateral. This secondary analysis indicated a similar relationship to that observed for bilateral hearing loss only, with attenuated effect sizes. However, this analysis assumed a constant causal effect across children with both lateralities of hearing loss, which we believe may not be reflective of the true behavior, and therefore have presented bilateral loss only within the main paper.

Our study design surveyed hearing amplification use at a single timepoint, which may miss capturing the variance in and accumulative effect of hearing amplification use over time [26]. All children with a permanent hearing loss who have Australian citizenship are eligible for hearing devices at no cost. We included children with no hearing device fitted in the never/rarely device use category for the aim 2 analysis. This approach may not be as relevant in populations with less generous and equitable access to hearing devices.

While born into the same geographic population, expressive vocabulary assessment occurred around a decade earlier for participants without known hearing loss compared to our DHH cohort. However, we believe our language performance comparisons remain valid since we do not expect typically developing children’s language outcomes to have changed greatly across time.

### 4.4. Interpretation in Light of Other Studies

Our findings support and extend findings from other studies by including children with unilateral loss and not limiting eligibility to those enrolled in EI programs. Like other existing studies of older children, we have demonstrated that children with even a mild degree of hearing loss have an expressive vocabulary below what is expected for their peers without hearing loss [27,28]. As also seen elsewhere [9,29], children with bilateral hearing loss demonstrated poorer expressive vocabulary as their degree of hearing loss increased. Slight delays for children with unilateral hearing loss align with reports for similar-age children with minimal hearing losses (unilateral and mild bilateral) [25,30].

We found no clear association between frequency of hearing amplification use and mean expressive vocabulary at age 2 years. The U-shaped performance curve where high vocabulary scores were found both for children reported to wear hearing amplification “never/rarely” and “always” warrants exploration. High hearing amplification use in children with high vocabulary scores may reflect the positive impact earlier detection through UNHS has on language outcomes [31] and the influence of hearing amplification use across time [29]. Alternatively, this group may represent a subgroup of children with hearing loss more likely to score highly in outcomes and adhere to hearing amplification use recommendations due to unmeasured factors. Reverse causation [32] cannot be ruled out for children with high vocabulary scores and low reported hearing amplification use, whereby parents/caregivers cease to enforce hearing aid use for children who are doing well and appear to be hearing without their amplification. This would also explain better expressive vocabulary scores in children with unamplified hearing loss at age 2 years.

There is some evidence that children without hearing loss with larger spoken expressive vocabularies at age 2 will, at age 5, show better performance in academic and behavioral outcomes compared to those with smaller spoken expressive vocabularies [33]. Early identification of spoken expressive vocabulary delays could personalize management, focusing on learning activities and experiences that help optimize skills for children with hearing loss across early childhood. Future research should explore underlying etiological or genetic markers that could predict language trajectories to direct resources to those DHH children in most need of intensive intervention.

## 5. Conclusions and Implications

This study provides population-level evidence of delayed spoken expressive vocabulary in DHH children at age 2 years, even with early detection through UNHS. We confirmed that the earlier the enrolment in EI programs and access to hearing amplification, the better the spoken expressive vocabulary outcomes. Specifically, our data supports the 1-2-3-month goal to screen, identify, and enter intervention rather than the original 1-3-6 EHDI benchmarks. While the extent of gains that could still be made is unknown, it is clear that interventions are not yet optimized for DHH children, as their early spoken expressive vocabulary outcomes are still poorer than their hearing peers. Moving forward, we must aim to more precisely understand the factors that impact language development in DHH children so that intervention can be targeted at those who need it most.

## Figures and Tables

**Figure 1 children-10-01223-f001:**
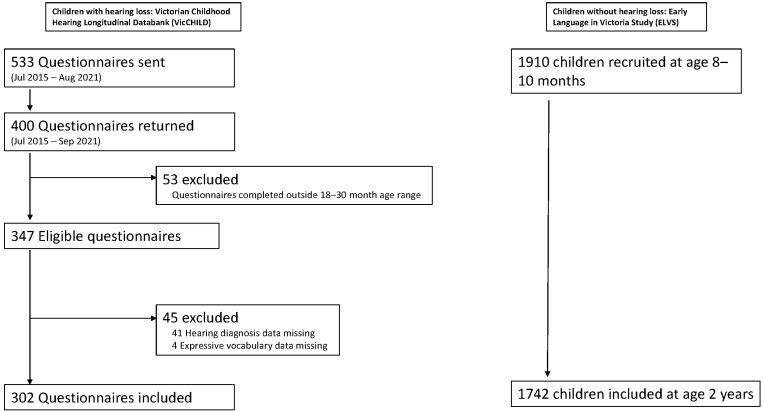
Participant flowchart.

**Figure 2 children-10-01223-f002:**
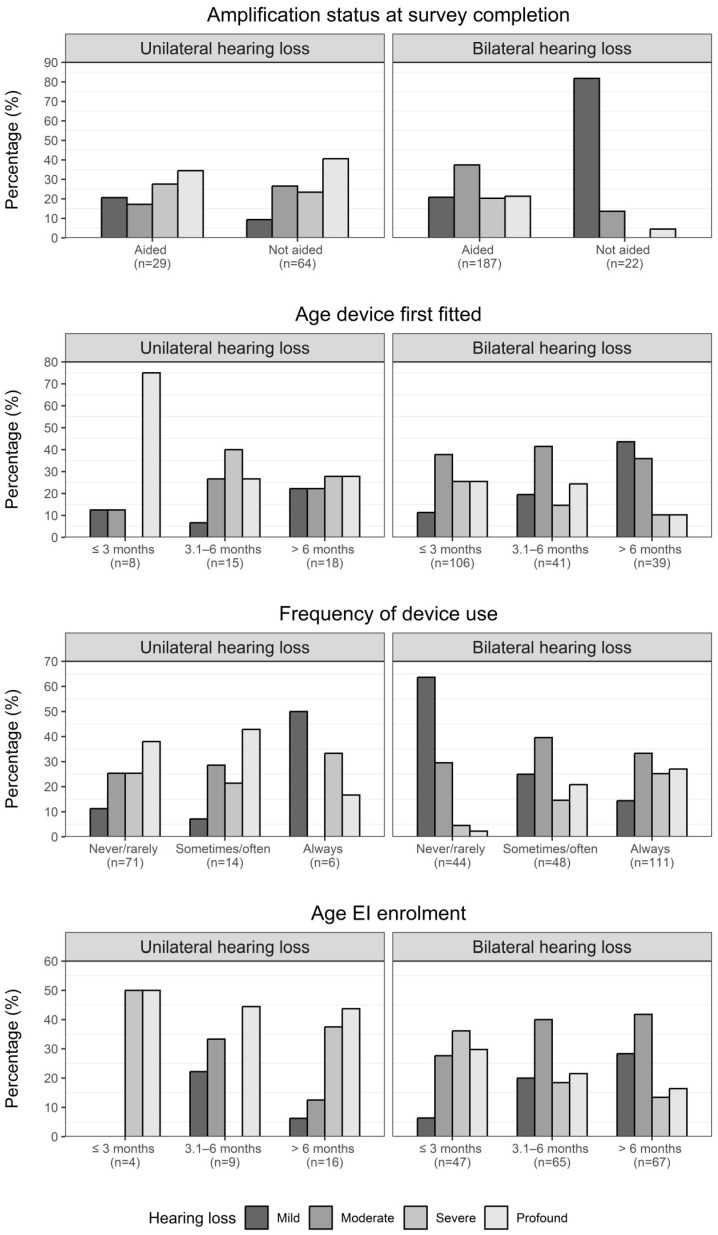
Hearing amplification and intervention characteristics of 2-year-old children broken down by unilateral and bilateral hearing loss. Percentages are calculated as the proportion of the degree of loss (mild to profound) within each *x*-axis group.

**Table 1 children-10-01223-t001:** Characteristics of the hearing loss and without hearing loss groups for aim 1.

				Hearing Loss Group (*n* = 302)		No Hearing Loss Group (*n* = 1742)	
				Missing ^a^ *n* (%)		Missing ^a^ *n* (%)	
**Demographic** **information**							
	Age at assessment, months—mean (SD)			0 (0)	25.4 (1.7)	0 (0)	24.2 (0.4)
	Sex: male—*n* (%)			0 (0)	170 (56.3)	0 (0)	887 (50.9)
	Parent’s education—*n* (%)			11 (3.6)		23 (1.3)	
			Year 10 or less		21 (7.2)		150 (8.7)
			Year 11		13 (4.5)		234 (13.6)
			Year 12		83 (28.5)		692 (40.3)
			Tertiary or Postgraduate		174 (59.8)		643 (37.4)
	Primary language at home—*n* (%)			12 (4.0)		8 (0.5)	
			English only		166 (57.2)		1644 (94.8)
			Bilingual (English and other language)		89 (30.7)		18 (1.0)
			Other language only		35 (12.1)		72 (4.2)
	Disadvantage Index, SEIFA—mean (SD)			1 (0.3)	1015.6 (61.1)	1 (0.1)	1037.6 (59.6)
**Birth-related factors**							
	NICU admission: yes—*n* (%)			7 (2.3)	50 (17.0)	0 (0)	284 (16.3)
	Gestational age, weeks—mean (SD)			1 (0.3)	38.3 (2.6)	152 (8.7)	39.3 (1.8)
**Hearing status—***n* (%)				0 (0)		0 (0)	
	Normal hearing				—		1742 (100.0)
	Unilateral hearing loss				93 (30.8)		—
		Mild			12 (4.0)		—
		Moderate			22 (7.3)		—
		Severe			23 (7.6)		—
		Profound			36 (11.9)		—
	Bilateral hearing loss				209 (69.2)		—
		Mild			57 (18.9)		—
		Moderate			73 (24.2)		—
		Severe			38 (12.6)		—
		Profound			41 (13.6)		—
**Outcome measures**							
	SSLM score, standardized—mean (SD)			0 (0)	88.5 (16.8)	0 (0)	98.7 (13.4)
	Combining words yet—*n* (%)			24 (8.0)		30 (1.7)	
		Not yet			81 (29.1)		290 (16.9)
		Sometimes			82 (29.5)		573 (33.5)
		Always			115 (41.4)		849 (49.6)

^a^ In the presence of missing data, the presented statistics are relative to records with available information for characteristics of interest.

**Table 2 children-10-01223-t002:** Estimates of the mean difference in standardized SSLM score within the hearing loss sample if all children had no hearing loss compared to if all children had hearing loss (broken down by severity of loss) under the three different models fitted.

		Unadjusted			Adjusted ^a^			Adjusted ^b^			
		Mean Difference	95% CI	*p*-Value	Adj. Mean Difference	95% CI	*p*-Value	Adj. Mean Difference	95% CI	*p*-Value	Adj. Mean ^b^
**Without hearing loss (ref)**		—	—	—	—	—	—	—	—	—	96.1
**Hearing loss: Unilateral**											
	Mild	−3.19	(−11.08, 4.69)	0.43	−3.43	(−24.27, 12.92)	0.73	−1.46	(−21.66, 13.46)	0.88	94.7
	Moderate	−8.19	(−14.56, −8.06)	<0.01	−4.12	(−11.75, 3.29)	0.28	−3.93	(−11.63, 3.47)	0.31	92.2
	Severe	−4.35	(−10.06, 1.37)	0.14	−2.17	(−9.57, 5.51)	0.57	−2.22	(−9.63, 5.24)	0.56	93.9
	Profound	−5.97	(−10.56, −1.39)	0.01	−4.50	(−10.42, 0.62)	0.11	−4.37	(−10.26, 0.71)	0.13	91.8
**Hearing loss: Bilateral**											
	Mild	−9.01	(−12.68, −5.34)	<0.01	−7.65	(−11.76, −2.90)	<0.01	−7.31	(−11.38, −2.48)	<0.01	88.8
	Moderate	−11.31	(−14.56, −8.06)	<0.01	−9.23	(−13.25, −4.76)	<0.01	−8.36	(−12.39, −4.20)	<0.01	87.8
	Severe	−15.98	(−20.45, −11.52)	<0.01	−13.51	(−17.79, −9.32)	<0.01	−13.35	(−17.78, −9.13)	<0.01	82.8
	Profound	−14.77	(−19.07, −10.46)	<0.01	−13.91	(−18.86, −8.80)	<0.01	−13.54	(−18.53, −8.37)	<0.01	82.6

^a^ Adjusted for the child’s sex, primary language at home, parent education, and SEIFA; ^b^ Further adjusted for NICU admission and gestational age.

**Table 3 children-10-01223-t003:** Hearing-related characteristics of children with hearing loss.

			Missing Information ^a^ *n* (%)	Hearing Loss Group (N = 302)	Unilateral Loss (N = 93)	Bilateral Loss (N = 209)
Age at diagnosis (months)			61 (30.2)			
	median (IQR)			1.0 (0.6, 1.6)	1.0 (0.7, 1.6)	0.8 (0.6, 1.4)
UNHS detected—*n* (%)			0 (0)	290 (96.0)	91 (97.8)	199 (95.2)
Any hearing device used ever—*n* (%)			0 (0)	231 (76.5)	41 (44.1)	190 (90.9)
Any hearing device used at survey—*n* (%)			0 (0)	216 (71.5)	29 (31.2)	187 (89.5)
	Hearing aid only			152 (50.3)	27 (29.0)	125 (59.8)
	Cochlear implant only			41 (13.6)	2 (2.2)	39 (18.7)
	Hearing aid and cochlear implant			23 (7.6)	0 (0)	23 (11.0)
	None			86 (28.5)	64 (68.8)	22 (10.5)
Age first hearing device fitted, continuous (months)			75 (24.8)			
	median (IQR)			3 (2, 6.5)	6 (4, 12)	3 (2, 6)
Age first hearing device fitted, categorical—*n* (%)			75 (24.8)			
	≤3 months			114 (50.2)	36 (45.0)	78 (53.6)
	3.1–6 months			56 (24.7)	23 (28.8)	33 (22.5)
	>6 months			57 (25.1)	21 (26.3)	36 (24.5)
Average hearing device use: week day—*n* (%)			8 (2.6)			
	Never/rarely, <4 h			115 (39.1)	71 (15.4)	44 (21.9)
	Sometimes/often, 4–8 h			62 (21.1)	14 (78.0)	48 (23.9)
	Always, >8 h			117 (39.8)	6 (6.6)	111 (55.2)
EI service used ever—*n* (%)			2 (0.7)	225 (75.0)	39 (41.9)	186 (89.9)
EI service used at survey—*n* (%)			78 (25.8)	209 (93.3)	29 (78.4)	180 (96.3)
Age EI program enrolment (months)			94 (31.1)			
		median (IQR)		6 (4, 10)	8 (5, 12)	6 (3, 9)
Age EI program enrolment—*n* (%)			94 (31.1)			
		≤3 months		51 (24.5)	4 (13.8)	47 (26.3)
		3.1–6 months		74 (35.6)	9 (31.0)	65 (36.3)
		>6 months		83 (39.9)	16 (55.2)	67 (37.4)

^a^ In the presence of missing data, presented values are relative to records with available information for characteristics of interest.

**Table 4 children-10-01223-t004:** Estimates of the mean difference in standardized SSLM score (for individuals with bilateral hearing loss) between intervention-related exposures: (a) hearing device used at survey; (b) average hearing device use; (c) average age of hearing device fitted; and (d) age at EI nrolment. Estimates were obtained under three different models.

		Unadjusted			Adjusted ^a^			Adjusted ^b^				
		Mean Difference	95% CI	*p*-Value	Adj. Mean Difference	95% CI	*p*-Value	Adj. Mean Difference	95% CI	*p*-Value	*n* *	Adj. Mean ^b^
**Hearing device used at survey**											199	
	No (ref)	—	—	—	—	—	—	—	—	—		90.6
	Yes	−2.81	(−10.16, 4.54)	0.45	−4.23	(−12.16, 3.71)	0.30	−5.36	(−13.21, 2.50)	0.18		85.2
**Age hearing device fitted**											177	
	<3 months (ref)	—	—	—	—	—	—	—	—	—		88.0
	3.1–6 months	−10.49	(−16.11, −4.87)	<0.01	−9.82	(−15.16, −4.47)	<0.01	−8.59	(−14.10, −3.08)	<0.01		79.4
	>6 months	−3.79	(−9.52, 1.93)	0.19	−5.66	(−11.39, 0.08)	0.05	−4.31	(−10.30, 1.69)	0.16		83.6
**Average hearing device use**											193	
	Always (ref)	—	—	—	—	—	—	—	—	—		86.1
	Sometimes/often	−5.75	(−11.32, −0.18)	0.04	−5.06	(−10.31, 0.20)	0.06	−4.62	(−9.80, 0.57)	0.08		81.4
	Never/rarely	2.55	(−3.20, 8.30)	0.38	3.27	(−3.07, 9.61)	0.31	3.39	(−2.86, 9.63)	0.29		89.4
**Age EI program enrolment**											171	
	<3 months (ref)	—	—	—	—	—	—	—	—	—		91.3
	3.1–6 months	−2.99	(−8.88, 2.90)	0.32	−5.60	(−11.11, −0.10)	0.05	−5.44	(−10.95, 0.07)	0.05		85.8
	>6 months	−8.92	(−14.78, −3.06)	<0.01	−10.59	(−16.28, −4.91)	<0.01	−9.95	(−15.68, −4.22)	<0.01		81.3

^a^ Adjusted for the child’s sex, primary language at home, parent education, SEIFA, severity of hearing loss; ^b^ Further adjusted for NICU admission and gestational age; * Number of records with complete information used for complete case analysis.

## Data Availability

All available data supporting the reported results can be found within this publication and its Supplementary Files. Data are not publicly available because not all VicCHILD and ELVS participants have provided consent for data sharing, and data sharing is limited to ethically approved research.

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
