# Peer review of "Spoken Expressive Vocabulary in 2-Year-Old Children with Hearing Loss: A Community Study"

_children, 2023, doi:10.3390/children10071223_

Round 1
Reviewer 1 Report
The authors give an overlook of the actual problem of speech expressive vocabulary development in 2-years old population. The study gives the opportunity to update data on this topic and highlights the need of early diagnosis of children with hearing loss.
Due to the complexity of the study, we suggest to use more figures in order to clarify methods and results.
Author Response
Thank you very much for the time taken to review our manuscript and for your suggestions. We tried to make sure there is a balance between presenting information in tables versus figures. For example, Figure 2 represents four separate sets of results, combined into a single figure for ease of viewing. The methodology for VicCHILD is published as a Data Resource Profile in the International Journal of Epidemiology (Sung V, Smith L, Poulakis Z, et al. Data Resource Profile: The Victorian Childhood Hearing Impairment Longitudinal Databank (VicCHILD). Int J Epidemiol. 2019;48(5):1409-1410h. doi:10.1093/ije/dyz168), there are many figures in that paper to represent our methodology, so we have not replicated them here in this manuscript. If the Editorial Team would like us to duplicate the figures, we are happy to consider obtaining permission from the IJE to do so. Thank you again for this suggestion.
Author Response
Thank you for your suggestions for our manuscript. Unfortunately, given the small number of participants at this young age who had cochlear implants (13.6%), this precluded us from conducting meaningful subgroup analyses. As a population-based study, we also had limited information on early intervention approaches like auditory-verbal therapy. This meant our analyses remained at a descriptive level for enrolment into any early intervention program, without detail on specific approaches.
Thank you for your suggestion about increasing the robustness of the discussion section. We have reported our paper according to a published suggestion of how to write the Discussion section ( https://www.icmje.org/recommendations/browse/manuscript-preparation/preparing-for-submission.html#f). Therefore, Section 4.1 is a summary of our results rather than a comparison with other studies. Section 4.4 is a discussion in comparison with other studies, we have tried to compare our study results with the most relevant existing publications.
We thank you for your suggested papers, and have added the following to Section 4.4:
New sentence added (lines 404-406) with two new references to support this statement:
‘Like other existing studies of older children, we have demonstrated children with even a mild degree of hearing loss have expressive vocabulary below what is expected for their peers without hearing loss.’
The sentence ‘Children with bilateral hearing loss demonstrated poorer expressive vocabulary as degree of hearing loss increased.’ has now been edited to enable a further two references to be cited, and now reads ‘As also seen elsewhere, children with bilateral hearing loss demonstrated poorer expressive vocabulary as degree of hearing loss increased.’
Thank you also for the suggestions on including the following papers:
https://pubmed.ncbi.nlm.nih.gov/32284003/ This paper is on cognitive and behavioural functioning in older deaf and hard of hearing children. While this is interesting, we do not feel this paper relates to our study aims, population or findings, and therefore we have not included it.
https://pubmed.ncbi.nlm.nih.gov/25758233/ This paper is a systematic review / meta-analysis on behavioural and emotional difficulties in older children and adolescents with hearing loss. While interesting, this paper does not relate to our study aims, population or findings, therefore we have not included it.
https://pubmed.ncbi.nlm.nih.gov/33369943/ This paper describes children’s emotion understanding in dynamic social situations using eye tracking. While interesting, this paper does not relate to our study aims, population or findings, therefore we have not included it.
Round 2
Reviewer 2 Report
Thanks for answering our questions, my congrats for your research.